A secure healthcare data transmission based on synchronization of fractional order chaotic systems

Suzelan Amir Nur Afiqah 1
http://orcid.org/0000-0001-9768-8291 Abd Latiff Fatin Nabila 2 3 fatinnabila@um.edu.my
Wong Kok Bin 1
Mior Othman Wan Ainun 1
1 Institute Science of Mathematics, Universiti Malaya , Petaling Jaya, Kuala Lumpur , Malaysia
2 Mathematics Division, Centre for Foundation Studies in Science, Universiti Malaya , Petaling Jaya, Kuala Lumpur , Malaysia
3 Digital Innovation & Smart Society Impact Lab, Taylor’s University , Subang Jaya, Selangor , Malaysia
Alatas Bilal
Electronic publication date: 2025 Feb 27
Publication date: 2025
Volume: 11
Electronic Location ID: e2665
Received 2024 May 22; Accepted 2025 Jan 2
Copyright: © 2025 Suzelan Amir et al.
Copyright year: 2025
Copyright holder: Suzelan Amir et al.
License: This is an open access article distributed under the terms of the Creative Commons Attribution License, which permits unrestricted use, distribution, reproduction and adaptation in any medium and for any purpose provided that it is properly attributed. For attribution, the original author(s), title, publication source (PeerJ Computer Science) and either DOI or URL of the article must be cited.
License URL: https://creativecommons.org/licenses/by/4.0/

Keywords: Healthcare system, Data transmission, Secure communication, Fractional-order chaotic systems

Funding: Taylor’s Internal Research Grant Scheme—Impact Lab Grant, Taylor’s University TIRGS-ILG/1/2023/SFE/003 Universiti Malaya BKP-ECRG-BKP042-2024-ECRG This work was supported by the Taylor’s Internal Research Grant Scheme—Impact Lab Grant, Taylor’s University (Grant no. TIRGS-ILG/1/2023/SFE/003). The APC was supported by the Universiti Malaya BKP-ECRG-BKP042-2024-ECRG. The funders had no role in study design, data collection and analysis, decision to publish, or preparation of the manuscript.

==============================
The transmission of healthcare data plays a vital role in cities worldwide, facilitating access to patient’s health information across healthcare systems and contributing to the enhancement of care services. Ensuring secure healthcare transmission requires that the transmitted data be reliable. However, verifying this reliability can potentially compromise patient privacy. Given the sensitive nature of health information, preserving privacy remains a paramount concern in healthcare systems. In this work, we present a novel secure communication scheme that leverages a chaos cryptosystem to address the critical concerns of reliability and privacy in healthcare data transmission. Chaos-based cryptosystems are particularly well-suited for such applications due to their inherent sensitivity to initial conditions, which significantly enhances resistance to adversarial violations. This property makes the chaos-based approach highly effective in ensuring the security of sensitive healthcare data. The proposed chaos cryptosystem in this work is built upon the synchronization of fractional-order chaotic systems with varying structures and orders. The synchronization between the primary system (PS) and the secondary system (SS) is achieved through the application of Lyapunov stability theory. For the encryption and decryption of sensitive healthcare data, the scheme employs the n-shift encryption principle. Furthermore, a detailed analysis of the key space was conducted to ensure the scheme’s robustness against potential attacks. Numerical simulations were also performed to validate the effectiveness of the proposed scheme.

Introduction

Healthcare institutions across various departments handle personally identifiable information (PII) and protected health information (PHI) through different health information systems. These systems, such as electronic health records (EHR), e-Prescribing software, remote patient monitoring systems, and laboratory information systems, are utilized by healthcare professionals including doctors, nurses, pharmacists, technicians, dietitians, and physical therapists. The regulatory framework governing PHI has undergone substantial transformations in recent decades. In the United States, the Health Insurance Portability and Accountability Act (HIPAA) was established in 1996 to ensure the protection of health information in terms of its usage, disclosure, storage, and transmission (Symatec, 2010; U. S. Department of Health and Human Services, 2018). In 2016, the European Union (EU) introduced the General Data Protection Regulation (GDPR) as a replacement for previous regulations, which came into effect in May 2018. The GDPR includes provisions and obligations regarding the handling of PII, including regulations on breach notification and penalties (European Union, 2016). While these regulations present challenges for healthcare institutions in terms of technology and data management, their main focus is to ensure data protection, enhance data security in hospitals, and ultimately preserve privacy of patient’s sensitive data.

To construct a secure data transmission in healthcare application, it is essential to address the security requirements of reliability and privacy. These two security goals are defined expressed as Amir, Malip & Othman (2020), Amir, Othman & Bin (2023): 1. For health data to be considered reliable, it must meet the requirements of both data integrity and data origin authentication. Data integrity: To guarantee the security of sensitive health data, measures are taken to prevent unauthorized modifications.

Data origin authentication: To verify the legitimacy of health data, it is essential to authenticate its origin from a legitimate entity.

2. User privacy is preserved when the conditions of anonymity and confidentiality are fulfilled. Anonymity: To protect the identity of a registered user on the system.

Confidentiality: To maintain strict access control and ensure that only authorized users can access the health data.

Security and privacy are critical concerns in advancing secure healthcare data transmission. The integration of technology in healthcare has significantly enhanced the precision and quality of services. However, there has been a noticeable rise in data violations within the healthcare industry since 2010, as reported by IBM in 2016 and the Ponemon Institute (2016). The compromised health data obtained from these violations raises serious concerns due to its immutability (Michael, 2017). Preserving data in the healthcare sector poses unique challenges due to the sensitive nature of the information involved and the potential impact on patient safety. Unlike credit card theft, where the bank can cancel the card, issue a new one, and provide reimbursement, the theft of PHI has long-lasting consequences. Personal details such as name, date of birth, insurance and healthcare provider information, as well as health and genetic data, are not easily modifiable. Once stolen, health information becomes highly valuable and can be exploited for various malicious activities, including identity theft and medical fraud. It is noteworthy that an individual’s health information holds more value on the dark web compared to their social security number or credit card details. It can be sold at significantly higher prices, ranging from 10 to 20 times more than other types of data (Humer & Finkle, 2014; Raul et al., 2016). This highlights the significant demand for health information and emphasizes the importance of robust data security measures within the healthcare industry.

Healthcare is an industry that heavily relies on the management of information, facilitated by the increasing adoption of information and communication technologies (ICTs). Although the current ICT tools in healthcare are considered to be at an early stage of development and may have certain limitations, there is a sense of optimism regarding the potential for significant advancements and improvements in numerous areas in the near future. One instance application of ICT in health care is EHR. EHR systems are comprehensive and continuous digital records that contain a patient’s health information over time. In the event of a compromise to the integrity of EHR or their encryption through an attack like ransomware, healthcare providers experience the loss of access and critical information essential for patient care. This includes vital details such as patient allergies, current medications, and co-existing medical conditions. Furthermore, cyberattacks have the potential to compromise the trust between doctors and patients, explicitly when there is a violation of data security (EPFL International Risk Governance Center, 2018). Cyberattacks pose a significant threat to hospitals, causing potential delays and disruptions in critical operations and jeopardizing patient safety. Examples of such incidents include the global WannaCry attack on the British National Health Service hospitals in May 2017 and the attack on the Hollywood Presbyterian Medical Center in February 2016. These attacks resulted in the postponement of surgeries and the need to redirect patients to nearby healthcare facilities (Ponemon Institute, 2016).

Considering the recent needs of secure healthcare data transmission, chaotic secure communication seen as a potential solution to provide a secure communication and privacy-preserving platform in transmitting data across healthcare communities. Chaotic secure communication systems exhibit a high degree of sensitivity, enabling the early detection of data violations. This sensitivity significantly enhances the system’s ability to thwart interception attempts, thereby safeguarding healthcare data from unauthorized access. In addition, chaotic secure communication offers several benefits, including cost-effectiveness, high confidentiality, and real-time signal processing (Wang, Zhong & Dong, 2016). To address the challenges in designing a secure healthcare transmission, we construct a new chaotic secure communication scheme where the underlying system is based on chaotic cryptosystem. The article presents a scheme that focuses on achieving synchronization between fractional-order chaotic systems of different structures and orders. The synchronization between the primary system and the secondary system is proven through the application of the Lyapunov stability theory. In addition, the scheme includes an encryption and decryption process for the main data signals, utilizing the n-shift encryption principle. The main contributions of this article can be summarized as: We have developed a generic model for data transmission in healthcare applications. This generic abstraction aims to provide a basis for future construction of secure data transmission protocol using synchronization of fractional order chaotic systems. To the best of our knowledge, this is the first comprehensive generic model for securely transmitting data in healthcare applications using fractional order chaotic systems.

We introduce a novel secure communication scheme that utilizes the synchronization of fractional-order chaotic systems with different orders. Notably, our design incorporates primary and secondary systems with varying structures and orders. This advanced scheme offers enhanced security performance and robustness.

We integrate the unique features of cloud computing with healthcare and devised a two-step encryption communication approach. In this method, the access node serves as a relay node for communication and incorporates timestamp-based encryption to enhance the key space. Furthermore, by harnessing the low latency and swift secondary capabilities of CC, this technique successfully fulfils the data transmission requirements of emergency computing and real-time communication within healthcare applications.

The structure of the remaining sections in this article is presented as: In “Literature Review”, we present a concise overview of cloud computing and the secure communication technology that utilizes chaotic synchronization. “Description of the Model” establishes the system model for our study. In “Chaotic Cryptosystem”, we present our proposed chaotic cryptosystem and provide the necessary preliminaries for understanding the underlying concepts and techniques. Moving on to “Our Secure Chaotic Secure Communication Systems”, we investigate into the exploration of the encryption and decryption processes of chaotic secure communication technology, along with the design of synchronization between two chaotic systems. We also provide a demonstration of the key space analysis. To assess the effectiveness of the proposed method, we present numerical simulations in “Security and Performance Evaluation”. Finally, in “Conclusion”, we summarize the findings of our research and discuss future research directions.

Literature review

Over the past decade, there has been a considerable amount of research focused on designing a secure communication in the field of healthcare. One notable contribution is presented in Chen et al. (2014), where a privacy authentication protocol was proposed for CC in the context of medical services. This protocol leveraged the characteristics of mobile devices, enabling convenient access to medical resources in the cloud for seeking medical advice. The authors claimed that their protocol provide robust security against common and typical attacks. However, subsequent studies raised several concerns about the effectiveness and security of the proposed protocol. Chiou, Ying & Liu (2016) conducted an evaluation and found that the protocol described in Chen et al. (2014) not only presented a high level of computational complexity but also failed to offer adequate patient privacy and message authentication. Building on this flaw, another research study Mohit et al. (2017) proposed that the protocol presented in Chiou, Ying & Liu (2016) still lacked the necessary measures to ensure both message authentication and patient privacy. In a separate study, Li, Shih & Wang (2018) identified further flaw in the protocol described in Mohit et al. (2017), highlighting its failure to protect patient privacy and maintain data unlinkability. Additionally, Liu & Ma (2018) discovered a significant vulnerability in the protocol proposed by Mohit et al. (2017), whereby any valid patient could easily obtain the private key of the cloud server. This discovery raised serious concerns as the knowledge of the private key gives a considerable security risk to the entire system.

Smart healthcare emerges as a new health revolution that integrates the intelligent technologies to provide patients with more efficient, flexible and personalized healthcare service (Hartmann, Hashmi & Imran, 2022; Liu & Tao, 2022; Shuo et al., 2019). However, the distribution of sensitive health data in the context of smart healthcare gives rise to numerous concerns regarding security and privacy (Algarni, 2019; Dawei & Guoquan, 2021; Noshina et al., 2020). Firstly, there is a need for a secure and efficient system to handle the vast amount of data transmitted within healthcare communities across various healthcare facilities. When relying on a single trusted server to manage the system, there is a potential risk of a single point of failure. If any part of the system fails, it can lead to a complete collapse and halt the operation of the entire system (Vikram, Aman & Vishal, 2021). This is particularly undesirable in a healthcare environment where regular updates on health data are essential. Secondly, health data is of utmost importance and highly sensitive, as it contains private information about patients. Protecting this data against unauthorized access is crucial. Disseminating health data poses a challenge, as it requires preserving patient confidentiality while ensuring that authorized parties have access to the necessary information.

A number of literatures uses blockchain to provide a privacy-preserving platform in disseminating data across healthcare communities (Hossein et al., 2021; Khatoon, 2020; Tanwar, Parekh & Evans, 2020; Shari & Malip, 2024). In the realm of blockchain technology, this approach employs a dynamic secret sharing mechanism based on the blockchain. These schemes have demonstrated enhancements in the security and success rate of data transmission. However, incorporating access control into the authentication scheme poses challenges, particularly in an industrial IoT environment (Hongwen et al., 2020). Additionally, the construction of a secure communication in healthcare should consider two main conflicting security goals of reliability and privacy (Amir, Malip & Othman, 2020; Amir, Othman & Bin, 2023). Data reliability is essential to accurately represent a patient’s health condition, ensuring that the data remains unaltered and originates from a legitimate source. Privacy is necessary to protect a user’s personal information from unauthorized access. However, achieving data reliability conflicts with the requirement for privacy, as verifying data reliability may reveal the sender’s information. Establishing a secure and efficient framework for data dissemination that balances these two security requirements is crucial to enable healthcare communities to fully leverage the benefits of the application.

The benefits of chaotic secure communication, including low cost, high security, and real-time signal processing, fit perfectly with the requirements of healthcare data transmission (Wang, Zhong & Dong, 2016). This technology has gained significant attention from academia and has become a popular research topic in recent years (An et al., 2011; Yang, Yang & Nie, 2014). For instance, Vaseghi, Pourmina & Mobayen (2017) investigated finite-time chaotic synchronization in the presence of noise and parameter uncertainties to achieve secure communication between the base station and sensor nodes using sliding mode control. Weiping et al. (2019) proposed a secure communication scheme based on the synchronization of memristive multidirectional associative memory chaotic neural networks, ensuring synchronization within a predetermined timeframe. Benkouider, Halimi & Bouden (2019) examined a secure chaotic communication system with discrete-time delay and introduced an encryption method using unknown input polytopic observers. Sham & Vidyarthi (2022) designed an efficient and intelligent method that classifies the applications arriving at the system into Class of Fog Applications (CoFA) and maps them to five levels of chaos-based security protocols for secure communication. However, it is important to note that these secure communication methods primarily focus on the synchronization of integer-order chaotic systems. Khan et al. (2022) focused on attentively observing, deftly retrieved the keyframe, and then processed the light cosine functions utilizing a hybrid method of chaotic map key frame picture encryption. From a security perspective, integer-order chaotic systems are vulnerable due to the exposure of initial values and model parameters.

Fractional-order chaotic systems have attracted significant attention in recent years due to their more intricate dynamic behavior when compared to integer chaotic systems (Megherbi et al., 2017). Fadia, Karim & Hamid (2019) have investigated secure communication schemes employing time-delayed fractional-order chaotic systems and validated their proposed electronic circuit through simulations using Multisim. Kiani-B et al. (2009) have developed a method to estimate the states of fractional chaotic systems using an extended fractional Kalman filter in the secondary system’s receiver. Additionally, a numerical simulation was conducted to compare the performance differences between an integer-order system and a fractional-order chaotic system. The utilization of fractional-order chaotic systems can improve security performance as their solutions exhibit high sensitivity to initial values. However, it is important to note that most existing studies have primarily focused on using primary and secondary systems with identical system and the same order of fractional-order chaotic systems. Wang, Huang & Zhao (2012) introduced a novel method for nonidentical synchronization of fractional-order Liu and Lorenz systems based on the Lyapunov stability theory. Abd Latiff & Mior Othman (2021) proposed the multi-fractional order of neural networks by multi-time delay (MFNNMD) approach to achieve stable chaotic synchronization, demonstrating the use of sliding mode control (SMC) based on time-delay chaotic systems. They incorporated the fractional-order Lyapunov direct method (FLDM) into SMC to maintain system stability and ensure global convergence of the error dynamics. Additionally, a high-security data protection protocol technique was introduced using a neural network system (Ettiyan & Geetha, 2023; Latiff & Othman, 2022). However, limited research has been conducted on synchronization techniques for fractional-order chaotic systems with different structures.

In another application discussed in Almuzaini & Abdullah (2023), the integration of chaos synchronization in healthcare applications is mentioned, however, there is no explicit presentation of security requirements to demonstrate the utilization of chaos synchronization. Xu & Ning (2023) introduced an innovative hybrid global adaptive coupling synchronization approach involving N Lorenz chaotic dynamical nodes. This method aims to establish a secure communication system between a base station and a multi-unmanned aerial vehicle (UAV) formation. The strategy incorporates feedback drive-response synchronization for the base station and the leader UAV, with UAVs connected via unidirectional adaptive coupling synchronization within the ad hoc network. Notably, the aspect of data reliability and privacy remains unaddressed (Xu & Ning, 2023). Additionally, Prakash & Manimegalai (2022) devised a robust and secure module to mitigate data loss in advanced technology systems. This module employs the Rossler, Lorenz oscillator, and Tent chaotic (RTL) maps to generate chaotic mask sequences. Nevertheless, the proposed scheme is vulnerable to potential attacks (Prakash & Manimegalai, 2022).

Based on the literature review, the schemes proposed in Chen et al. (2014), Chiou, Ying & Liu (2016), Liu & Ma (2018), Mohit et al. (2017), Almuzaini & Abdullah (2023), Prakash & Manimegalai (2022), Xu & Ning (2023) demonstrate a lack of comprehensive security measures, particularly in addressing the dual requirements of reliability and privacy. Additionally, the schemes in Hartmann, Hashmi & Imran (2022), Liu & Tao (2022), Shuo et al. (2019), Hossein et al. (2021), Khatoon (2020), Tanwar, Parekh & Evans (2020), Shari & Malip (2024) integrate blockchain technology with smart healthcare systems. However, there remains a degree of skepticism within the industry concerning the security of blockchain technology itself (Hongwen et al., 2020). In parallel, chaotic secure communication has been explored in works such as Weiping et al. (2019), Vaseghi, Pourmina & Mobayen (2017), Benkouider, Halimi & Bouden (2019), Sham & Vidyarthi (2022), Khan et al. (2022) which utilize integer-order chaotic systems. However, integer-order chaotic systems are less secure due to their lower complexity. Meanwhile, fractional-order chaotic systems have been analyzed in Megherbi et al. (2017), Kiani-B et al. (2009), Wang, Huang & Zhao (2012), Abd Latiff & Mior Othman (2021), Xu & Ning (2023). However, none of these works have applied fractional-order chaotic systems in the context of secure healthcare data transmission. Motivated by this gap, we have developed a secure and efficient framework for healthcare data dissemination, which addresses the critical requirements of both reliability and privacy. Our approach leverages fractional-order chaotic systems, which offer enhanced security due to their sensitivity to initial conditions, thereby providing a more robust solution for secure healthcare applications. Comparative analysis of state-of-art for secure data transmission is tabulated in Table 1.

Table 1 Comparative analysis of state-of-the-art for secure data transmission.

Ref	Year	Application	Objectives	Limitations	
Benkouider, Halimi & Bouden (2019)	2019	Chaotic time varying delayed system	Study secure chaotic communication with delays, using input polytopic observers.	Emphasizes synchronization of integer-order chaotic systems.	
Fadia, Karim & Hamid (2019)	2019	Fractional order delayed chaotic system	Explore secure communication with fractional-order and time delays, validated via Multisim.	Focus on identical primary and secondary systems.	
Weiping et al. (2019)	2019	Fixed time synchronization control	Design secure communication using synchronized chaotic neural networks.	Not suitable for healthcare applications.	
Tanwar, Parekh & Evans (2020)	2020	Blockchain in smart healthcare	Enable varied access levels for patient data.	No chaos synchronization protocol model.	
Khatoon (2020)	2020	Blockchain in smart healthcare	Regulate patient information sharing among healthcare organizations.	No chaos synchronization mechanism suggested.	
Hossein et al. (2021)	2021	Blockchain in smart healthcare	Empower patients to control access to their data.	Security and privacy concerns.	
Prakash & Manimegalai (2022)	2022	Chaotic algorithms (Rossler, Lorenz, Tent)	Generate chaotic mask sequences for secure communication.	Method susceptible to attacks.	
Sham & Vidyarthi (2022)	2022	Chaotic algorithms (Chua)	An adaptive sliding mode security scheme is developed using Chua’s chaotic in CoFA framework.	Processing speed is quite large.	
Xu & Ning (2023)	2023	Chaotic system with cubic non-linear term	Propose a new 3D chaotic system, analyze dynamics, circuit implementation, and signal flow.	Lacks presentation of security requirements.	
Almuzaini & Abdullah (2023)	2023	Hybrid synchronization in UAV communication	Secure communication between main station and UAVs.	Data reliability and privacy issues.	
Shari & Malip (2024)	2024	Blockchain in smart healthcare	Utilize Sign-Proxy and enhanced DPoS for a secure consortium blockchain.	The integration of signcryption, proxy re-encryption, and DPoS significantly complicates system design and implementation.	
Our work	2025	Fractional order chaotic systems	Introduce secure communication for healthcare using different fractional-order chaotic systems.	–	

Description of the model

System model

The data transmission architecture composed a user, cloud server, access point (AP) and doctors. The secure channel is represented by dotted arrows, while the public channel is represented by solid arrows. The responsibilities of each entity are outlined below: Users. These entities, each with their own unique identities, act as the primary participants in the system. They possess and manage one or more smart IoT devices that are connected to an access point (AP). They conveniently input their personal medical health data to the AP using wireless terminals, whether they are at home or outside.

Access point (AP). The access point acts as a relay node for communication and has certain computing and storage capacity. The incorporation of the access point enables timely nodes to time-sensitive tasks, ensuring real-time interactions. Additionally, AP transmit extensive data to the cloud server for long-term storage, facilitating comprehensive analysis by doctors for further examination.

Cloud server. The cloud server is characterized by its high performance and substantial storage capacity, offering superior computational power and extensive storage capabilities. The cloud is responsible for verifying the trustworthiness of data. This work operates under the assumption that the cloud server is a reliable and trustworthy entity.

Doctor. These group of professional entities will utilize the received data and address any anomalies or medical needs. In addition, they able to obtain the correct diagnosis, which has the potential to reduce the necessity for hospitalization and can even save lives during medical emergencies.

Data transmission architecture

The data transmission architecture in a healthcare cloud involves four main phases: Encryption, Transmission, Securing, and Decryption. In the Encryption phase, a fractional-order chaotic system acts as the primary system, generating three chaotic signals as encryption keys. During the Transmission phase, health data is shared between the user and doctor via an access point (AP), which serves as a gateway. Data are encrypted using chaotic signals through masking modulation and are transmitted securely using n-shift encryption. In the Securing phase, the AP creates a time-based matrix function for dual encryption using a randomly generated matrix and a timestamp, which also doubles as a new encryption key. The encrypted data are then sent to the doctor for analysis and to the cloud for storage. Finally, in the Decryption phase, the doctor uses an identical random matrix generator and timestamp key provided by the AP to decrypt the data. The fractional-order chaotic system is reconstructed as a secondary system to generate three chaotic signals as decryption keys. The different phase can be summarized and depicted in Fig. 1.

Figure 1 Generic model.

Chaotic cryptosystem

This section provides definitions for three types of fractional order derivatives found in the literature: Riemann-Liouville, Grunwald-Letnikov, and Caputo. Among these definitions, it is widely accepted that only the Caputo definition maintains the same form as integer-order differential equations, particularly when taking initial conditions into account (Zhen, Xia & Zhao, 2012).

Definition 1 (see Abd Latiff & Mior Othman, 2021) The Caputo fractional derivative of order α for a continuous function (F∈Cg([t0,+∞],R)) for Γ is the Gamma function is described as given in Eq. (1)

(1) DαF(t)={1Γ(g−α)∫t0tFg(τ)(t−τ)α−g+1dτ,ifg−1<α<g,1Γ(1−α)∫t0tF′(τ)(t−τ)αdτ,if0<α<1.wheret≥t0,g∈Z+,Γ(Z)=∫0∞tZ−1e−tdt.

We present our chaotic transmitter and chaotic receiver, respectively. Table 2 shows some of the notations used in our chaotic system for ease of reading throughout the article.

Table 2 Table of the symbol and notation.

Symbol	Notation	
α,β	Derivative order	
p,q,r,u,v,w	System parameter	
(x˙1)(t),(x˙2)(t),(x˙3)(t)		
(Y˙1)(t),(Y˙2)(t),(Y˙3)(t)	Chaotic signal	
c1,c2,c3	Controllers	
ts1	Timestamp	

Definition 2. The fractional order Lorenz system is defined as given in Eq. (2) (Petráš, 2011).

(2) {Dα(x˙1)(t)=p(x˙1)(t)−p(x˙2)(t)Dα(x˙2)(t)=q(x˙1)(t)−(x˙2)(t)−(x˙1)(t)(x˙3)(t)Dα(x˙3)(t)=(x˙1)(t)(x˙3)(t)−r(x˙3)(t).

The chaotic signals in the sending node x˙1(t),x˙2(t),x˙3(t) are generated using the Lorenz system. The system in Eq. (2) displays chaotic behavior when α=0.9941 and p=10, q=28 and r=83 (see Petráš, 2011; Bo, Zhong & Dong, 2016).

Definition 3. The fractional order Rössler system is defined as given in Eq. (3) (Petráš, 2011).

(3) {Dβ(Y˙1)(t)=−Y˙2(t)−(Y˙3)(t)+c1Dβ(Y˙2)(t)=−(Y˙1)(t)+u(Y˙2)(t)+c2Dβ(Y˙3)(t)=v+(Y˙3)(t)(Y˙2)(t)−w(Y˙3)(t)+c3.

The chaotic signals in the sending node Y˙1(t),Y˙2(t),Y˙3(t) are generated using the Rössler’s system. The system in Eq. (3) displays chaotic behavior when β=0.9 and u=0.5 and v=0.2 (see Petráš, 2011; Bo, Zhong & Dong, 2016).

Our secure chaotic secure communication systems

Proposed model

The system consists of four entities: the users, access point, cloud, and doctors. To establish secure communication, the user source node functions as a chaotic transmitter. By utilizing fractional-order chaotic systems, known for their robust nonlinearity and sensitivity to initial conditions, the security of the system is enhanced compared to integer-order chaotic systems. The primary system (PS) in this scheme is the classical fractional-order Lorenz system, which generates three chaotic signals as encryption keys. The safety health data message, denoted as m(t), is encrypted using chaotic encryption. It is modulated with one of the state variables of the (PS) and undergoes n-shift encryption with another state variable. To add an extra layer of security, the encrypted data is transmitted to the access point instead of directly to the doctor node. At the access point, a random matrix generator creates a function matrix with time as a variable, further encrypting the data. This random matrix, along with the timestamp and a new key, enhances the security of the encryption. The access point then forwards the encrypted data to the doctor node through the cloud layer. On the receiving end, the doctor node acts as the chaotic receiver in the secure communication. The secondary system (SS) is implemented using the fractional-order Rössler system, which generates three chaotic signals as decryption keys. It is important to note that the encryption key used in the data encryption (PS) is different from the decryption key used in the (SS). Synchronization between the two state variables acting as keys is essential. Unlike traditional secure communication schemes, this approach employs chaotic systems with different structures for data encryption and decryption, making it more challenging for potential attackers. Chaotic encryption and synchronization serve as the foundational application in this secure communication scheme. To decrypt the data, the doctor node is equipped with an identical random matrix generator as the access point. The collaborative timestamp key is employed for data decryption at the receiving node.

Chaotic generation and encryption

In the following steps, a user (U) establishes communication with the doctor (D) through a secure channel via the access point (AP). The fundamental principle of chaotic secure communication relies on utilizing a nonlinear chaotic system as a source of broadband pseudo-random signals. These signals are employed to encrypt the original message, resulting in a meaningless signal that is then transmitted through a shared communication channel.

Step ①: By selecting one of the state variables, denoted as x˙2(t), generated by the primary system given in Eq. (2), and combining it with the main signal mn(t), we obtain a masked signal ms(t)=x˙2(t)+mn(t). This masked signal is then subjected to n-shift encryption.

Step ②: The key signal, denoted as k1(t), is derived from one of the state variables of the primary system given in Eq. (2). The n-shift cipher is defined using this key signal as in Eq. (4).

(4) n−shift(ms(t))=a[…a[a[ms(t),k1(t)],k1(t)]…,k1(t)]=Enc(t).

where the parameter c is selected in such a way that the signals ms(t) and key1(t) are constrained to the range (−c,c), while still maintaining the nonlinearity of the function f(x˙2(t),k1(t)) depicted in Eq. (5).

(5) f(x˙2(t),k1(t))={(x˙j(t),k1(t))+2c,if2c≤(x˙j(t),k1(t))≤−c(x˙j(t),k1(t)),if−c≤(x˙j(t),k1(t))≤c(x˙j(t),k1(t))+2c,ifc≤(x˙j(t),k1(t))≤2c.

Step ③: The encrypted signal, denoted as Enc(t), is transmitted to the access point. In the access point, a random matrix generator generates a random function matrix where time is considered as an independent variable. For a fixed time, it is a constant matrix as shown in Eq. (6).

(6) B(t)=(b11(t)b12(t)…b1m(t)b21(t)b22(t)…b2m(t)⋮⋮⋱⋮bm1(t)bm2(t)…bmm(t))m×m.

Chaotic data transmission

In this phase, user (U) generates a health data and sends it to doctor (D) through access point (AP). The following information provides further details on this process:

Step ④: The access point selects a timestamp ts1 by randomly choosing a number ts1∈{0,1}R. associated with the ongoing data transmission and generates a constant matrix B1(t) through a random selection process. The ts1 is to ensure the freshness of the message during data transmission.

Step ⑤: Upon generating a random matrix, the encryption for the signal Enc(t) for secondary encryption is performed as described in Eq. (7):

(7) 2Enc(t)=B1(ts1)×Enc(t).

Securing the chaotic data

After receiving the message, the cloud executes the following procedures:

Step ⑥: The receiver, which utilizes chaotic dynamics, is also capable of generating random matrices with a similar structure.

Step ⑦: In this scenario, the access point only transmits encrypted signals, denoted as 2Enc(t), along with a timestamp, represented as ts1 which ensures key freshness and synchronization.

Step ⑧: The (AP) disseminates the safety data M to the doctor via the cloud.

Step ⑨: By utilizing the timestamp key ts1 during decryption, the encrypted signals 2Enc(t) can be transformed using the matrix B(t) to obtain as in Eq. (8):

(8) Enc(t)=B1−1(ts1)×2Enc(t)

Chaotic decryption

Step ⑩: The decryption process follows the same rule as the encryption process depicted in Eq. (9).

(9) Decryptms(t)=d(Enc(t))n-shift(ms(t))=a[…a[a[Enc(t),−key^1(t)],−key^1(t)],…,−key^1(t)].

The decryption key key^1(t) is used in the secondary system and needs to be accurately matched to key1(t) in order to decrypt the encrypted signal.

Step ⑪: The original main signal mn(t) can be recovered by as depicted in Eq. (10)

(10) mn(t)=ms(t)−Y˙2(t).

Synchronization

In this section, we focus on achieving synchronization between the fractional-order chaotic primary system (PS) and the secondary system (SS). We define an error system that captures the differences between the two systems and transform the synchronization problem into ensuring the stability of the trivial solution of the error system. To analyze this stability, we employ Lyapunov stability theory, specifically adapted for fractional-order dynamical systems. High synchronization errors indicate a significant misalignment between the chaotic systems, which negatively impacts the decryption process. Accurate synchronization is crucial for successful decryption, as any deviation in synchronization can lead to incorrect or compromised recovery of the original message.

For the synchronization of chaotic systems given in Eqs. (2) and (3), the synchronization errors are defined as Petráš (2011) given in Eq. (11)

(11) ei(t)=Y˙i(t)−x˙i(t)fori=1,2,3.

Definition 4 (see Petráš, 2011) The equation given in Eqs. (2) and (3) are said to achieve synchronization if there exist controllers c1,c2,c3 such that asymptotically synchronization error system shown in Eq. (12)

(12) limt→∞⁡||ei(t)||=0wherei=1,2,3.

Assume a given autonomous system as given in Eq. (13)

(13) Dαx(t)=∑j=1naijxjwherei=1,2,3.andDαistheCaputofractionalderivativeoforderα

Definition 5 (see Petráš, 2011) The trivial solution of autonomous system given in Eq. (12) is said to be asymptotically stable if and only if

limt→∞||xi(t)||=0wherei=1,2,3.

Lemma 1 (see Bo, Zhong & Dong, 2016) The trivial solution of autonomous system given in Eq. (11) is asymptotically stable if and only if as in Eq. (14)

(14) |arg(eig(A))|>απ2whereA=(aij)n×n.

Key space analysis

The utilization of chaotic systems in secure information transmission is embedded in their unique properties, such as sensitivity to initial conditions and highly nonlinear behavior. These characteristics make chaotic systems particularly suitable for ensuring the confidentiality and integrity of transmitted data. Traditional secure communication schemes often adopt the same chaotic system for both the primary system (PS) and the secondary system (SS). In such schemes, the key space of an integer-order chaotic system is limited to the selection of initial value conditions and system structure.

To illustrate this limitation as shown in Eq. (15), let us consider the example of a three-dimensional Rössler chaotic system. In the conventional approach, the initial value conditions of both the (PS) and (SS) are chosen as parameters in determining the encryption and decryption keys.

(15) Forinitialvalue=0Key={(x1,x2,x3),(x1)′,(x2)′,(x3)′,(a,b,c,)}.

where a,b,c are the three parameters that determine the structure of the chaotic Rössler system.

The existing secure communication scheme, characterized by a key space of nine parameters, is not able to meet the security requirements in healthcare cloud development. Recall that, this limited key space, as denoted by equation given in Eq. (15), is inadequate to ensure robust protection of sensitive healthcare data. To address this limitation, the present contribution introduces a novel scheme with an extended key space, offering a more comprehensive set of parameters.

By expanding the key space, the proposed scheme aims to enhance the security measures employed in healthcare cloud environments as presented in Eq. (16).

(16) Forinitialvalue=0Key={(x˙1,x˙2,x˙3),α,(p,q,r),(Y˙1,Y˙2,Y˙3),β,(u,v,w),ts1}.

Security and performance evaluation

Security analysis

This section highlights security analysis. The presence of adversaries in the network is a common assumption in vehicular communications (Anushruthika & Justus, 2024; George & Kamara, 2022). In the context of chaos-based cryptography, which is often applied in secure communications, the security depends on system configurations, key length, and sensitivity to initial conditions. The chaotic system is highly sensitive to parameter changes, meaning that even slight variations can result in entirely different outcomes, enhancing security.

Robustness against attack

Robustness against Brute force attack

Claim 1. Our work resilient against brute force attack

Proof. This key space consists of 15 parameters. If each parameter is represented by a double-digit number accurate to 10 decimal places.

By calculating the total number of key spaces using given in Eq. (16) representation, denoted by Nkey given in Eq. (17),

(17) Nkey=1010×…×1010=10150=2150log210

The total number of key spaces is greater than or equal to 10150 given in Eq. (17). This indicates a vast number of possible key combinations, providing strong resistance against brute force attacks. Therefore, our fractional-order chaos synchronization scheme, with such a large key space, meets the necessary security requirements for transmitting sensitive healthcare data, ensuring strong protection against brute force cryptanalysis.

Robustness against man in the middle (MITM) attack

Claim 2. Our work robust against MITM attack

Proof. The fractional-order chaos synchronization method used in our scheme is highly sensitive to initial conditions and system parameters. Any slight alteration in the transmitted signal would cause a significant divergence in the receiver’s synchronization process, making it easy to detect tampering attempts by an external attacker. This sensitivity ensures that any modification introduced by an adversary would result in failure of the synchronization, alerting both communicating parties.

In addition, The vast key space 10150 further enhances security against MITM attacks. Since the attacker cannot feasibly guess or intercept the correct keys due to the sheer number of possibilities, the system is resistant to impersonation or replay attacks often seen in MITM scenarios.

Robustness against replay attack

Claim 3. Our work robust against replay attack

Proof. The unpredictability and inherent complexity of chaotic systems present significant challenges for attackers who attempt to predict or reconstruct the encryption process. The chaotic dynamics ensure that the output of the system remains highly non-repetitive and nearly impossible to reproduce, thereby providing robust security against efforts to reuse intercepted ciphertext.

In some chaotic encryption methodologies, time-variant signals or sequences are integrated into the encryption process. These dynamic components evolve over time, ensuring that each encryption operation generates a unique result. Replay attacks are rendered ineffective because the system relies on a specific time-related context that the replayed data does not provide. In this context, the access point transmits only encrypted signals, denoted 2Enc(t), alongside a timestamp, represented as ts1. Inclusion of the timestamp ts1 improves the overall security of communication and guarantees the freshness of the transmitted data.

Numerical simulation

In this section, we conducted simulations to demonstrate the effectiveness and feasibility of the proposed scheme. The simulations were performed on a Linux machine running an Intel Core i5-4,790 processor at a frequency of 3.6 GHz. MATLAB has been used to illustrate numerical simulations with sampling time St=0.005. A sampling time of St=0.005 provides a good resolution for observing the behavior of the system. The simulation starts at 0 s and runs for a total of 100 s to observe the chaotic behavior of the system. The primary system is modeled using the Lorenz system, while the secondary system utilizes the Rössler System. The summary simulation of this protocol was carried out using the following design parameters in Table 3.

Table 3 Simulation configuration parameters.

Parameter	Description	Value	
Time interval	Simulation duration	0–100 s	
Sampling time	Time step for numerical solution	0.005	
Initial conditions	Starting values for state variables	[1.5, 0.1, 0.1]	
Fractional order	Order of the fractional derivative	0.775≤α≤0.995	
Control gain (K)	Gains for active controller	[10.3, 0.3, 10.3]	
γ	Parameter for synchronization adjustment	100.05	
Noise level	Amplitude of added external noise	0.01	

The simulation results of the attractors of the Lorenz system (Petráš, 2011) and the Rössler system (Petráš, 2011; Bo, Zhong & Dong, 2016) are shown in Fig. 2. The trajectories in a three-dimensional state space were plotted for the given parameters: σ=10, ρ=28, β=83, order α=0.775 for Fig. 2, order α=0.855 for Fig. 2, order α=0.995 for Fig. 2, with initial conditions (x1(0),x2(0),x3(0))=(1.5,0.1,0.1). These trajectories provide a visual representation of the system’s behavior and demonstrate the evolution of the system’s variables over time. Notably, the systems also exhibit chaotic behavior for orders within the range of 0.775≤α≤0.995, demonstrating the persistence of chaotic behaviour across various fractional orders.

Figure 2 Attractors synchronization is shown for different orders in (A) 0.775, (B) 0.885 and (C) 0.995.

Simulation without control input

Simulation without control input refers to a scenario where a system is simulated without applying any external controls or inputs that would typically influence its behavior. Equations (2) and (3) without the controller was solved for the σ=20, ρ=58, β=83, order α=0.955, and initial conditions (x1(0),x2(0),x3(0))=(1.5,0.1,0.1). Figure 3 depict the synchronization of variable states between the secondary and primary systems according to the model studied. It is apparent that the primary according to the three state components depicted in Fig. 3. The curves of state variables xi for i = 1, 2, 3 are represented by blue lines, and the curves of state variables yi for i = 1, 2, 3 are denoted by red lines. Figure 3 display the synchronization errors without control input.

Figure 3 Synchronization without control input dynamics of system (2) and (3) with respect to the variable (A) x1, y1, (B) x2, y2, (C) x3, y3 and (D) shows the without control input scenario.

These trajectories provide a visual representation of the system’s behavior and demonstrate the evolution of the system’s variables over time. Notably, the systems also exhibit chaotic behavior for orders within the range of 0.775≤α≤0.995, demonstrating the persistence of chaotic behaviour across various fractional orders. The plot of the x-axis and y-axis highlights the differences between the Lorenz and Rössler systems, which arise from their distinct chaotic dynamics. The Lorenz system exhibits rapid oscillations with larger amplitudes, while the Rössler system starts with smaller oscillations that grow over time. Initially, the values are large due to significant differences in their trajectories, leading to spikes and oscillations. As synchronization progresses, the values decrease but may not converge to zero because of inherent structural and dynamical differences between the systems, as well as numerical precision limitations.

Simulation with control input

Simulation with control input involves studying how a system responds to external influences or commands applied to it. By incorporating control inputs, we can assess the system’s ability to adapt to changes, manage disturbances, and achieve specific performance goals. Incorporating control input is a well-established approach in chaos synchronization and is commonly assumed in numerous studies (Abd Latiff & Mior Othman, 2021; Latiff & Othman, 2022). The systems described by Eqs. (2) and (3) along with the controller were numerically solved for σ=20, ρ=48, β=83, order α=0.950, and initial conditions (x1(0),x2(0),x3(0))=(10.3,0.3,10.3), considering the existing chaotic attractors. Figure 4 depict the states of the variables in a synchronized manner between the primary and secondary systems based on the study’s model that converge with control input to zero. Solutions of Eqs. (2) and (3) were obtained to demonstrate chaotic synchronization within a short time period t. The curves of state variables xi for i = 1, 2, 3 are represented by blue lines, and the curves of state variables yi for i = 1, 2, 3 are denoted by red lines. The synchronization errors are graphed in Fig. 4, and it shows from these figures that synchronization error is converge to zero relatively quick.

Figure 4 Synchronization dynamics with control input for systems (2) and (3) are shown for: (A) variables x1 and y1, (B) variables x2 and y2, (C) variables x3 and y3, and (D) synchronization error with control input.

These trajectories provide a visual representation of the system’s behavior and demonstrate the evolution of the system’s variables over time. Notably, the systems also exhibit chaotic behavior for orders within the range of 0.775≤α≤0.995, demonstrating the persistence of chaotic behaviour across various fractional orders. The plot of the x-axis and y-axis demonstrates the effect of external input on synchronization between the Lorenz and Rössler systems. With the external input applied, the values of the synchronization errors gradually decrease and converge to zero over time. This convergence indicates effective synchronization despite the structural and dynamical differences between the systems. The external input compensates for these differences, aligning the trajectories of the two chaotic systems.

Chaos synchronization in healthcare application

Comparison of synchronization errors between integer-order of Lorenz system and Rössler system against fractional-order of system Lorenz system and Rössler system

In this subsection, we analyze and compare the synchronization performance of integer-order (Mata-Machuca et al., 2012; Bendoukha, Abdelmalek & Ouannas, 2019) and fractional-order systems, using the Lorenz system as the primary and the Rössler system as the secondary. The goal is to assess the evolution of synchronization errors over time for both integer-order and fractional-order systems. Through this comparison, we highlight the advantages of fractional-order systems, particularly their faster convergence and more stable synchronization, which are essential for secure communication applications, as shown in Figs. 5 and 6. Additionally, we extract synchronization error values between 9.75 and 10.0 s, which are tabulated in Table 4.

Figure 5 Synchronization error for Lorenz system.

Figure 6 Synchronization error for Rössler system.

Table 4 Synchronization errors for Lorenz and Rössler systems at different time from 9.75–10.0 s.

Time (s)	Lorenz system	Rössler system	
	e1	e2	e3	e1	e2	e3	
9.75	5.1949	3.9785	−7.1255	−4.8954	−4.5664	39.489	
9.755	5.0714	3.7825	−7.4236	−5.0384	−4.7771	39.172	
9.76	4.9404	3.5735	−7.7081	−5.1902	−5.0017	38.871	
9.765	4.8012	3.3513	−7.9783	−5.3516	−5.2404	38.587	
9.77	4.6531	3.1156	−8.2335	−5.523	−5.4934	38.32	
9.775	4.496	2.8656	−8.4725	−5.7046	−5.7611	38.071	
9.78	4.3293	2.601	−8.6943	−5.8968	−6.044	37.842	
9.785	4.1527	2.3213	−8.8977	−6.1	−6.3422	37.633	
9.79	3.9657	2.0262	−9.0814	−6.3144	−6.6561	37.447	
9.795	3.768	1.7156	−9.2439	−6.5404	−6.9856	37.284	
9.8	3.559	1.3892	−9.3838	−6.7784	−7.3306	37.146	
9.805	3.3382	1.0471	−9.4995	−7.0288	−7.6912	37.035	
9.81	3.105	0.68944	−9.5893	−7.2921	−8.0669	36.951	
9.815	2.859	0.31645	−9.6514	−7.5687	−8.4575	36.898	
9.82	2.6	−0.071261	−9.6844	−7.8585	−8.8621	36.877	
9.825	2.3279	−0.47331	−9.6861	−8.1616	−9.2803	36.889	
9.83	2.0426	−0.88929	−9.6536	−8.4781	−9.7115	36.937	
9.835	1.7439	−1.3183	−9.5843	−8.8078	−10.155	37.024	
9.84	1.4321	−1.7592	−9.4757	−9.1505	−10.609	37.152	
9.845	1.1075	−2.2101	−9.3256	−9.5058	−11.071	37.323	
9.85	0.77043	−2.6689	−9.1316	−9.8731	−11.541	37.54	
9.855	0.42163	−3.133	−8.8917	−10.252	−12.014	37.804	
9.86	0.061852	−3.5994	−8.6041	−10.641	−12.488	38.117	
9.865	−0.30793	−4.0646	−8.267	−11.039	−12.959	38.481	
9.87	−0.6866	−4.5246	−7.8788	−11.445	−13.423	38.898	
9.875	−1.0728	−4.9751	−7.438	−11.858	−13.877	39.367	
9.88	−1.4652	−5.4118	−6.9435	−12.277	−14.314	39.891	
9.885	−1.8613	−5.8304	−6.3955	−12.697	−14.732	40.469	
9.89	−2.2584	−6.2239	−5.7936	−13.118	−15.123	41.1	
9.895	−2.6537	−6.5854	−5.1388	−13.536	−15.48	41.785	
9.9	−3.0438	−6.9083	−4.4328	−13.947	−15.797	42.521	
9.905	−3.4254	−7.1862	−3.6786	−14.349	−16.067	43.305	
9.91	−3.795	−7.4132	−2.8803	−14.737	−16.284	44.132	
9.915	−4.1485	−7.5836	−2.0432	−15.108	−16.443	44.998	
9.92	−4.482	−7.692	−1.1735	−15.457	−16.539	45.895	
9.925	−4.7913	−7.7334	−0.27873	−15.78	−16.566	46.816	
9.93	−5.0717	−7.703	0.63267	−16.074	−16.52	47.753	
9.935	−5.3195	−7.5962	1.552	−16.333	−16.396	48.696	
9.94	−5.5316	−7.4104	2.4699	−16.555	−16.191	49.636	
9.945	−5.7028	−7.1471	3.3738	−16.735	−15.909	50.56	
9.95	−5.8288	−6.8088	4.2516	−16.868	−15.55	51.457	
9.955	−5.9062	−6.3989	5.0926	−16.951	−15.118	52.315	
9.96	−5.9326	−5.9217	5.8868	−16.982	−14.619	53.124	
9.965	−5.9067	−5.3823	6.6256	−16.96	−14.056	53.876	
9.97	−5.8279	−4.7867	7.3011	−16.883	−13.437	54.562	
9.975	−5.6969	−4.1419	7.9066	−16.754	−12.769	55.176	
9.98	−5.5151	−3.4556	8.4362	−16.572	−12.059	55.711	
9.985	−5.285	−2.7365	8.8854	−16.341	−11.315	56.164	
9.99	−5.0087	−1.9956	9.2506	−16.063	−10.55	56.531	
9.995	−4.6878	−1.2455	9.5303	−15.74	−9.7763	56.81	
10.0	−4.326	−0.49596	9.7252	−15.376	−9.0033	57.002	

The results demonstrate that fractional-order synchronization between the Lorenz primary system and the Rossler secondary system outperforms integer-order synchronization in terms of stability and accuracy. Although both systems eventually achieve synchronization, the fractional-order approach shows faster convergence and more stable synchronization errors over time, indicating its superior performance in chaotic communication systems. This enhanced behavior highlights the advantages of fractional-order systems in secure data transmission, where faster and more reliable synchronization is critical for encryption and decryption processes.

Application in secure healthcare data transmission

Chaotic communication systems typically involve the utilization of two chaotic oscillators: one functioning as the transmitter and the other as the receiver. The message is concealed within the chaotic signal’s noise, which the transmitter sends. The results of the novel chaotic synchronization system follow a straightforward approach to facilitate secure transmission of healthcare data. The primary system (2) operates as the transmitting system, referred to as PS, while the secondary system (3), denoted as SS, serves as the receiving system.

In subsequent numerical simulations, the system’s characteristics and the initial conditions for both the transmitter and receiver systems are assumed to adhere to the specifications outlined in “Our Secure Chaotic Secure Communication Systems”. Figure 7 provides a visual representation of a numerical simulation showcasing the practical application of chaotic synchronization in secure healthcare data transmission, illustrating the original, masked and recovered messages. The trajectories in a three-dimensional state space were plotted for the given parameters: σ=10, ρ=28, β=83, order α=0.961, and initial conditions (x1(0),x2(0),x3(0)),x4(0),x5(0),x6(0)=(1.0,1.0,1.0,2.0,2.0,1.0). The selection of a 10-s time interval highlights the emphasis on demonstrating the system’s speed and efficiency in decrypting data.

Figure 7 Illustration of the secure transmission process: (A) the original transmitted message, (B) the masked message for security, and (C) the recovered message after decryption.

Conclusion

In conclusion, the conventional healthcare system faces significant challenges in appointment scheduling, leading to limited availability, missed appointments, and increased waiting times for treatments. To address the reliability and privacy concerns related to data transmission in the healthcare application, we have proposed a novel secure communication scheme based on chaos cryptosystem. By utilizing the synchronization of fractional-order chaotic systems with different structures and orders, we ensure enhanced security for data transmission. The synchronization between the primary system (PS) and the secondary system (SS) is established using the Lyapunov stability theory, providing a robust foundation for secure communication. The n-shift encryption principle is employed for encrypting and decrypting the safety health data, ensuring confidentiality and data integrity throughout the transmission process. We have thoroughly analyzed the key space of our proposed scheme, demonstrating its ability to resist brute force attacks and meet the stringent security requirements of healthcare data transmission. Additionally, through numerical simulations, we have showcased the effectiveness of our scheme in ensuring reliable and secure communication in the healthcare domain.

In conclusion, our novel approach based on chaos synchronization provides a promising solution to the challenges faced by the conventional healthcare system. By incorporating cutting-edge techniques from chaos cryptosystem and fractional-order chaotic systems, we can enhance the security and efficiency of data transmission in healthcare applications. Our research contributes to the growing body of knowledge in secure communication for healthcare, paving the way for more effective and reliable healthcare services worldwide. The limitation indicates that the current work is concentrated on a limited range of parameters and applications, and further research is required to evaluate its effectiveness and reliability across a wider range of scenarios. For future research, it would be valuable to explore implementations using hybrid orders and to apply the proposed protocol in a range of applications, including vehicular communication, while ensuring the security requirements are not compromised.

Supplemental Information

Supplemental Information 1 Synchronization with system (1) and (2) fde12.

Supplemental Information 2 Synchronization with system (1) and (2) parameters.

Additional Information and Declarations

Competing Interests

The authors declare that they have no competing interests.

Author Contributions

Nur Afiqah Suzelan Amir conceived and designed the experiments, performed the experiments, analyzed the data, performed the computation work, prepared figures and/or tables, authored or reviewed drafts of the article, and approved the final draft.

Fatin Nabila Abd Latiff conceived and designed the experiments, performed the experiments, analyzed the data, performed the computation work, prepared figures and/or tables, authored or reviewed drafts of the article, and approved the final draft.

Kok Bin Wong conceived and designed the experiments, authored or reviewed drafts of the article, and approved the final draft.

Wan Ainun Mior Othman conceived and designed the experiments, authored or reviewed drafts of the article, and approved the final draft.

Data Availability

The following information was supplied regarding data availability:

The simulation code are available in the Supplemental Files.

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
