# Peer review of "A secure healthcare data transmission based on synchronization of fractional order chaotic systems"

_PeerJ Computer Science, doi:10.7717/peerj-cs.2665_

## Round 0.1 · original submission · Major Revisions

Dear authors,

Thank you for the submission. The reviewers’ comments are now available. It is not suggested that your article be published in its current format. We do, however, advise you to revise the paper in light of the reviewers’ comments and concerns before resubmitting it. Furthermore, equations should be used with correct equation number. Please do not use “as follows”, “given as”, etc. Explanation of the equations should also be checked. All variables should be written in italic as in the equations. Their definitions and boundaries should be defined. Necessary references should be provided. Many of the equations are part of the related sentences. Attention is needed for correct sentence formation.

Best wishes,

·

Basic reporting

Needs massive improvement in writing professionsl English to improve the clarity of the concepts. Currently, a number of key concelts are still ambiguous.
Graphs for the results need improvemnet for better visibility. The graphs showing similar content can be grouped in one graph.

Experimental design

Research questions are not well defined. It also lacks in providing sufficient knowledge on how the gap is filled.
Methods are not defined with sufficient details about the key concepts.

Validity of the findings

Impact and novelty not assessed.
Underlying data is not provided and in-depth clarification of results.
Secondly, the justification for the use/benefits of controlled and un-controlled input is not explicitly explianed.
Results are not compared with the dominating base scheme to claim the dominance of this work.

Additional comments

After line 379, How the m and n are defined to calculate the matric B(t)

After line 387, the 2Enc(t) involves the multiplication with matrix B(ts1) with Enc(t). How to ensure the order of matrix B(t) and Enc(t) which is mandatory for the possiblity of multiplication. As the Enc(t) is genarted through hashes, then how van it be multiplied with a matrix of order m x n.

In lines 394-395, "By incorporating the timestamp ts1
as a key, the security of the communication is enhanced". It is not verified that the attacjet can also generate the timestamp to decrypt the message.

After line 408, how the subtraction of Y2(t) can generate mn(t). X2(t) was added to get ms(t). It is not verifid wether Y2(t) is same as X2(t) or its reverse. It must be explicitly clarified in description.

The synchronization errors is calculated as ei(t) = Y
i(t) = xi(t). If the error is high then, will the decryption process be successful or not?? Section 4.6 needs careful revison with technical support to address this point.

The state variables are used but the exact values and iys types are not defned for better clatity of this concept.

Moreover, a number of concepts are used without defining or citing the reference to view those details.
Cite the figure numbers in "The simulation results are shown in Figure ??. Figure ??"

Reviewer 2 ·

Basic reporting

Need little modification as mentioned in additional comments

Experimental design

Need little modification as mentioned in additional comments

Validity of the findings

Need little modification as mentioned in additional comments

Additional comments

The author discusses a method to introduce a novel secure communication scheme that utilizes chaos cryptosystem to address the reliability and privacy concerns related to data transmission for healthcare application. The proposed chaos cryptosystem in our work is based on the synchronization of fractional-order chaotic systems with varying structures and orders. The synchronization between the master system (MS) and the slave system (SS) is established using the Lyapunov stability theory. To encrypt and decrypt the safety health data, our scheme employs the n-shift encryption principle. We calculated and analyzed the key space of the scheme to ensure its security. Additionally, numerical simulations are conducted to demonstrate the effectiveness of the proposed scheme.
The article is looks good and work done in a proper manner but here are some suggestion to enhance this paper for the better article and further it can be accepted:
1. Try to reduce complex sentences that mostly used in abstract, Introduction and other section throughout article.
2. Include findings and result in abstract which can more attract towards readers.
3. Missing Figure Number at line 456 The simulation results are shown in Figure ??. Figure ?? depicts the attractors of the Lorenz system and the Ro¨ssler system.
4. Add and compare to more suitable healthcare data encryption article such as https://onlinelibrary.wiley.com/doi/full/10.1155/2022/8853448, https://content.iospress.com/articles/journal-of-intelligent-and-fuzzy-systems/ifs201770

5. If possible, add some security analysis and show effectiveness your proposed work.
6. Rephrase line 510 “In conclusion, our novel approach based on chaos synchronization provides a promising solution to the challenges faced by the conventional healthcare system.” Should be deleted In conclusion,

Reviewer 3 ·

Basic reporting

- overall, the article was clearly written, however, it did not highlight the research gap and most importantly, failed to highlight what was the main issues/challenges in healthcare date transmission and justify why a new secure communication scheme is needed compare to existing secure communication scheme between health systems.
- the abstract appears to be rather disconnected when it starts with describing challenges in healthcare appointment and then jumped to proposing secure data transfer. how does appointment scheduling relates to secure data transfer? abstract should be concise and highlight the main problem, gap, what the paper proposing, and its results/contribution.
- introduction section did not provide enough background to the problem, mainly due to the problem statement in this paper is not clear.
- The literature review didn’t provide enough discussion on what is the gap in existing secure healthcare data transmission model/solutions and how the novel chaotic secure communication system address that gap and different than other similar fractional order chaotic based system (to better highlight the novelty)
- The proposed solution contribution is generic enough and not specifically unique to healthcare data transmission.
- literature references acceptable, but missing more recent relevant articles related to fractional order chaotic model.
- article structure contains a couple of errors such as figures are located within list of references and raw data not shared (its unclear how to generate the simulated data using the provided code)
- figures need to be more descriptive.

Experimental design

- The description of the model is too high level by only showing cloud-ap-user-doctor as the entity? how the proposed model could play a role in the context of health information system architecture (since the author claims the model is designed for health data transmission.)
- most importantly, numeric simulation is too generic and does not appear to be related to health data. The simulation is also not well explained and unclear what is the objective of each simulation, especially with regards to health data transmission.
- the numerical simulation also contains some error such as unknown Figure ?? and its unclear what system (1) and system (2) refers to.

Validity of the findings

- the numerical simulation results also lacks of comparison with other approach to show
- overall, the finding reported is not derived by using heath data and thus we cant validate the claim that the proposed model is design to address the gap in securing health data transmission model.

Additional comments

The work presented in the paper may have merits, however, it is not properly written to highlight the research gap, poor design of experiment, lack of useful discussion on the results, and most importantly, failed to highlight what was the main issues/challenges in healthcare date transmission and justify why a new secure communication scheme is needed compare to existing secure communication scheme between health systems. Based this itself, the paper requires rigorous rewrite.

---

## Round 0.2 · Major Revisions

Dear Authors,

I am grateful for your efforts in revising the paper. Only one of the prior reviewers was available to re-review, so we sought an additional reviewer to also comment.

According to Reviewer 1 the manuscript still needs improvement. Please address the comments of Reviewer 1 and resubmit your paper.

Best wishes,

·

Basic reporting

Still needs improvement in writing professional English to improve the clarity of the concepts.
Currently, a number of key concepts are still ambiguous and technical details are missing

Experimental design

Simulation paramers are not well defined and main metrics are also not explored well

Validity of the findings

Still, the justification for the use/benefits of controlled and un-controlled input is not explicitly
explained. Results are not compared with the dominating base scheme to claim the dominance of this work.

Additional comments

1) Equation numbers are not cited properly in the paragraph above that equation.
2) References are not cited for even very main level equations. These equations must have been taken from some other research papers that are mandatory to cite for avoiding plagiarism.
3) After line 378, the 2Enc(t) involves the multiplication with matrix B(ts1) with Enc(t). It is still not clear in the manuscript, how to ensure the order of matrix B(t) and Enc(t) which is mandatory for the possibility of multiplication. If the Enc(t) generates a Vector then how to ensure its order to multiply with matrix of order m x n.
4) Line 384, "By incorporating the timestamp ts1 as a key” is still the same as in previous version. Is it a typo or the ts1 is used as a secret key for encrypting the data?
5) A few replies that clarify the concept should also be appended in the manuscript for the clarity of the readers as well. Append the text in Section 4.6 as replied to my last comment as “High synchronization errors suggest a significant misalignment between the chaotic systems, which adversely affects the decryption process. Accurate synchronization is critical for successful decryption, as deviations in synchronization can lead to incorrect or compromised recovery of the original message.”
6) In simulation parameters, add the configuration parameters for conducting the simulation like different parameters of the equations that can directly affect the performance and similar parameters. Do not write MATLAB, Master and salve system instead discuss in paragraph only.
7) In Section 4.2, h function is not defined which is used in Equation (3). It seems to be hash as usually used but it is mentioned in reply to my previous comment that it is not hash. In simulation parameters, h is also used for sampling time. It would be better to use a different variable to represent the function and also describe its role.
8) Line 495, 502-503 describe the values used for the parameters and what are its impact if values are varied. Also explore why these values are chosen for simulation. In Section 4.2, r=8/3 was also used. Is it the same used for Beta?
9) The results are segregated in without / with control input to zero. It is not explicitly stated Why these two scenarios are important? What is the main impact that the results through controlled input is more synchronized?
10) For all the graphs, explicitly describe why a value is small or large for the Master and Slave systems. Also discuss an example value from the graph and justify the reason of high or low values.
11) Figure 12 to 14 is also not described well. Provide technical justification of these results and which values are plotted. Why ti from 0 to 10 whereas in other graphs, it is from 0 to 100. Also mention the unit at the end of titles for x and y axes.
12) In reply to my comment “Results are not compared with the other base scheme to claim the dominance of this work”, it is mentioned that results are compared with base schemes but I could not found any. I could see the results for Master (classical fractional-order Lorenz system) for encryption and Slave (fractional-order Rossler system) for decryption. These results are not compared with any other existing scheme discussed in literature to analyze the performance of proposed scheme. In Section 5.3, include the references of the base schemes (if any discussed in literature) that are used to compare the results with the proposed work.
13) In contributions, it is claimed that “our design incorporates master and slave systems with varying structures and orders” but the results do not explore any variation in structure and order. There may be different set of results if structures and orders are varied during simulation.
14) In conclusion, include the limitations of this work before writing future work.

Reviewer 4 ·

Basic reporting

The paper proposes a new secure communication scheme specifically for healthcare data transmission, utilizing chaos-based cryptosystems. This approach focuses on protecting sensitive health information from unauthorized access while ensuring the reliability of data. The study develops a unique secure transmission model using fractional-order chaotic systems, which are more secure than integer-order systems due to higher sensitivity to initial conditions and system parameters. This sensitivity enhances resistance against various cyberattacks.The synchronization between a master system and a slave system is achieved through the application of Lyapunov stability theory, supporting robust encryption and decryption processes.The encryption uses an n-shift principle and a timestamp-based encryption layer, increasing the key space significantly to prevent brute force attacks. The key space includes fractional-order parameters and other variables, making the scheme highly resilient to interception attempts.
The language is generally clear and formal, effectively conveying the technical details.The paper provides a thorough background on secure data transmission, referencing foundational works and recent studies in chaotic systems, cryptography, and healthcare data security. Mathematical proofs, key space analysis, and simulations are provided to validate the robustness of the proposed model.

Experimental design

The experiments were implemented on MATLAB, run on a Linux machine with specific parameters such as a sampling time h=0.005 and initial conditions. The simulations used the Lorenz system as the master and the Rössler system as the slave, both of which are well-known chaotic systems. This setup ensures reproducibility as the parameters and initial conditions are explicitly definedaotic System Dynamics.The authors identify a gap in applying fractional-order chaotic systems specifically for healthcare data security, which is generally addressed through more common encryption techniques or integer-order chaotic systems. This focus on fractional-order systems fills a knowledge gap by enhancing security sensitivity through chaotic synchronization. The investigation is conducted rigorously, with detailed simulations in MATLAB to verify the efficacy and security of the chaotic model for healthcare data.

Validity of the findings

The use of fractional-order chaotic systems is based on solid theoretical foundations. By leveraging the inherent sensitivity and unpredictability of chaotic systems, the authors provide a robust justification for their approach to secure healthcare data transmission. The findings are supported by MATLAB-based simulations, which demonstrate the chaotic system’s capacity to mask, transmit, and recover healthcare data securely. Numerical results and visualizations (such as the chaotic attractors and synchronization plots) validate that the chaotic systems are functioning as intended. The study includes a key space analysis to demonstrate robustness against brute-force attacks, contributing to the validity of the encryption strength.

Additional comments

All revisions requested by the reviewers after the first review process have been thoroughly addressed and incorporated into the manuscript. Each comment was carefully considered to enhance the accuracy and clarity of the study, and all suggested corrections have been fully implemented

---

## Round 0.3 · Minor Revisions

Dear Authors,

According to Reviewer 1 the manuscript still needs improvement. Please address the minor comments of Reviewer 1 and resubmit your paper.

Best wishes,

·

Basic reporting

Simulation parameters are not well defined and main metrics are also not explored well
Needs further explanations for the metrics in the results Section

Experimental design

Little corrections suggested

Validity of the findings

Needs further ecploration as mentioned in the comments

Additional comments

1) Line 384, "By incorporating the timestamp ts1 as a key” is still the same as in previous version. Is it a typo or the ts1 is used as a secret key for encrypting the data?
In the reply, it is stated as “: In this scenario, the access point only transmits encrypted signals, denoted as 2

Reviewer 2 ·

Basic reporting

Satisfied with this updated manuscript

Experimental design

Satisfied with this updated manuscript

Validity of the findings

Satisfied with this updated manuscript

Additional comments

The updated manuscript is now well-edited and incorporated various comments. from my side, it can be acceptable.

---

## Round 0.4 · Minor Revisions

Dear Authors,

According to Reviewer 1 the manuscript still needs improvement. Please address the comments of Reviewer 1 and resubmit your paper. If these comments are not addressed after the fourth revision, your paper may be rejected for publication.

Best wishes,

·

Basic reporting

My last time comments were not shown properly. Only a half of first comment was shown in report. I am attaching again as PDF as well.
Simulation parameters are not well defined and main metrics are also not explored well
Needs further explanations for the metrics in the results Section

Experimental design

Little corrections suggested

Validity of the findings

Needs further ecploration as mentioned in the comments

Additional comments

1) Line 384, "By incorporating the timestamp ts1 as a key” is still the same as in previous version. Is it a typo or the ts1 is used as a secret key for encrypting the data?
In the reply, it is stated as “: In this scenario, the access point only transmits encrypted signals, denoted as 2 Enc (t), along with a timestamp, represented as ts 1, serving as an additional key.”
It is fine to say that the “encrypted signal contains a timestamp” but it cannot be stated as a key because the entire message is encrypted with a key to generate an encrypted signal. Time stamp is only used for ensuring the key freshness while the message is decrypted at receiver side. It may happen that the ts can be a part to calculate the secret key but it cannot be used as a key. Time stamp can easily be guessed from the transmission time which is a publically known value.
2) Not addressed. In simulation parameters, add the configuration parameters for conducting the simulation like different parameters of the equations that can directly affect the performance and similar parameters. In Table 3, no need to write MATLAB, Master and salve system instead discuss these in paragraph above the Table.
3) Not addressed. Figure 12, 13 and 14 also lacks in describing technical justification of results. In x-axis, why the values of t are from 0 to 10 whereas in other graphs, it is from 0 to 100. Also mention the unit at the end of titles for x and y axes.
4) The results compared in the Master (classical fractional-order Lorenz system) for encryption and Slave (fractional-order Rossler system) for decryption should also be presented in your work.

Minor Corrections:
5) For the justification to the without / with control input to zero, also add this text in the manuscript from the line 501 onwards.
6) Figure 1 can be removed. Section 2.2 can be reduced by adjusting in one paragraph as the information is already stated in a similar way in figure 2.2.
7) Equation numbers are still not mentioned in the text above that equation e.g. writing as “given in equation (1).”
8) In reply to comment 10, “For all the graphs, explicitly describe why a value is small or large for the Master and Slave systems”. The technical description mentioned in the reply should also be written in the manuscript.
9) In Section 5.3, cite the reference # of the base schemes (master and Slave systems).

---

## Round 0.5 · accepted · Accept

Dear Authors,

Thank you for addressing the reviewers' comments. Your paper seems sufficiently improved. Please correct minor edits suggested by Reviewer 1 in production phase.

Best wishes,

·

Basic reporting

little formatting points provided, rest is fine.

Experimental design

Improved as per my comments

Validity of the findings

Improved as per my comments

Additional comments

Minor Corrections related to formatting:
1) Mention the unit at the end of titles for x and y axes for all graphs where applicable
2) Captions of the Tables are placed at the top of Table. Please follow as per the standard for the Journal given in template.
3) Caption of Figure 2,3,4 can be merged with one title at bottom of figures as “Figure 2. Attractors synchronization is shown for different orders in (a) 0.775, (b) 0.885 and (c) 0.995”. Use (a), (b) and (c) below each figure. Similarly, Figure 5,6,7,8 can be merged with one main Title as “Figure 3. Synchronization without control input dynamics of system (2) and 3) with respect to the variable (a) x1, y1, (a) x2, y2, (a) x3, y3 and (d) shows the without control input scenario.”. Similarly, correct for Figure 9 - 12 and for Figure 15-17 as well. Correct the citation of figure number in the paragraphs above to these figures as well.